# Neuroprotection in Glaucoma: NAD^+^/NADH Redox State as a Potential Biomarker and Therapeutic Target

**DOI:** 10.3390/cells10061402

**Published:** 2021-06-05

**Authors:** Bledi Petriti, Pete A. Williams, Gerassimos Lascaratos, Kai-Yin Chau, David F. Garway-Heath

**Affiliations:** 1NIHR Biomedical Research Centre, Moorfields Eye Hospital and UCL Institute of Ophthalmology, London EC1V 9EL, UK; b.petriti@ucl.ac.uk; 2Department of Clinical & Movement Neurosciences, UCL Queens Square Institute of Neurology, London NW3 2PF, UK; k.chau@ucl.ac.uk; 3Department of Clinical Neuroscience, Division of Eye and Vision, St. Erik Eye Hospital, Karolinska Institutet, 171 64 Stockholm, Sweden; Pete.williams@ki.se; 4King’s College Hospital NHS Foundation Trust, London and King’s College London, London SE5 9RS, UK; gerassimos.lascaratos@nhs.net

**Keywords:** glaucoma, mitochondrial dysfunction, retinal ganglion cell (RGC), nicotinamide adenine dinucleotide (NAD^+^), NAD^+^/NADH redox state, ATP, neurodegenerative disease

## Abstract

Glaucoma is the leading cause of irreversible blindness worldwide. Its prevalence and incidence increase exponentially with age and the level of intraocular pressure (IOP). IOP reduction is currently the only therapeutic modality shown to slow glaucoma progression. However, patients still lose vision despite best treatment, suggesting that other factors confer susceptibility. Several studies indicate that mitochondrial function may underlie both susceptibility and resistance to developing glaucoma. Mitochondria meet high energy demand, in the form of ATP, that is required for the maintenance of optimum retinal ganglion cell (RGC) function. Reduced nicotinamide adenine dinucleotide (NAD^+^) levels have been closely correlated to mitochondrial dysfunction and have been implicated in several neurodegenerative diseases including glaucoma. NAD^+^ is at the centre of various metabolic reactions culminating in ATP production—essential for RGC function. In this review we present various pathways that influence the NAD^+^(H) redox state, affecting mitochondrial function and making RGCs susceptible to degeneration. Such disruptions of the NAD^+^(H) redox state are generalised and not solely induced in RGCs because of high IOP. This places the NAD^+^(H) redox state as a potential systemic biomarker for glaucoma susceptibility and progression; a hypothesis which may be tested in clinical trials and then translated to clinical practice.

## 1. Introduction

Glaucoma is one of the most common neurodegenerative diseases and the leading cause of irreversible blindness worldwide. Its prevalence increases with age and affects ~80 million people worldwide, with primary open–angle glaucoma (POAG) being the most frequent form. Due to the rapid increase in ageing populations worldwide, it is estimated that the number affected will increase to 111.8 million in 2040 [1]. The number of hospital related glaucoma visits surpasses one million per year in England and Wales alone, putting a significant strain on health services [2]. The disease is often associated with fear of vision loss, consequent social withdrawal, and depression from impaired vision causing a significant psychological burden to the patient as vision decreases [3]. Thus, glaucoma is a significant social and economic burden. This underlines the need to prioritise research in this area and to develop new treatments for glaucoma.

Progressive neurodegeneration of retinal ganglion cells (RGCs; the output neurons of the retina) and their axons, which make up the optic nerve, is the hallmark of glaucoma. The optic nerve is particularly sensitive to mitochondrial dysfunction and bioenergetic failure; mitochondrial dysfunction plays a causative role in the disease pathogenesis of optic neuropathies such as Leber’s hereditary optic neuropathy (LHON) and autosomal dominant optic atrophy (ADOA). Mitochondrial dysfunction has been demonstrated to play a significant role in the neurodegenerative cascade of RGCs in glaucoma. Whilst the significance of mitochondrial involvement in neurodegeneration is well established, the underlying mechanisms remain unclear, especially in glaucoma. Mitochondrial dysfunction is associated with numerous age–related neurodegenerative diseases including Parkinson’s disease (PD) and Alzheimer’s disease (AD) and it is likely that these diseases may share common pathophysiological mechanisms [4,5].

Various pathways directly or indirectly associated with pathologic changes in mitochondrial metabolism have been implicated in mitochondrial dysfunction and ultimately neuronal cell death. Changes related to nicotinamide adenine dinucleotide (NAD) metabolism have been proposed to play a significant role in neurodegeneration [6]. NAD^+^, and its reduced form NADH, are essential cofactors in redox metabolism and signalling in all forms of cellular life. They are central to metabolic pathways such as glycolysis, the tricarboxylic acid (TCA) cycle and oxidative phosphorylation (OXPHOS) [7]. These pathways ultimately culminate in the production of adenosine triphosphate (ATP) which is the energy currency of all cells. Therefore, the NAD^+^(H) redox state is of utmost importance for energy (ATP) generation, required for action potential transmission through the axons of the RGCs. In addition, NAD^+^ is key to activities of NAD^+^–consuming enzymes such as sirtuins, poly–ADP–ribose polymerases (PARPs) and cyclic ADP–ribose synthases (cADPRs; CD38 and CD157), thereby implicating NAD^+^ in cellular processes such as cell signalling, DNA repair, cell division, ageing and epigenetics [8].

## 2. Glaucoma and Mitochondrial Function

Intraocular pressure (IOP) is a main risk factor for glaucoma, however, a large proportion of patients develop optic neuropathy with IOP in the statistically normal range (normal tension glaucoma; NTG) or continue to progress despite IOP–lowering treatment. At present all current treatments for glaucoma are for IOP–lowering and, while lowering IOP can be beneficial to slow progression, an important subset of patients with NTG and high tension glaucoma (HTG) still lose significant vision despite treatment [9]. A study in around 600 POAG patients followed from diagnosis to death found a prevalence of 42% of blindness in one eye and 16% in both eyes at their last visit [10]. In addition, many patients have IOPs above the norm and never progress to visual dysfunction suggesting that raised IOP cannot be the sole defining factor in glaucoma and, therefore, other factors must confer susceptibility in glaucoma development and progression. These factors may be linked to multiple damage pathways, such as vascular (e.g., Flammer syndrome [11]) and biomechanical optic nerve head weakness [12], including those influencing mitochondrial function. Numerous neurodegenerative disorders have been linked to a decline in the mitochondrial electron transport chain (ETC) activity [13]. The optic nerve has one of the highest oxygen consumption rates and energy demands of any tissue in the body demonstrated by presence of large numbers of mitochondria in the RGCs [14]. In fact, the unmyelinated portion of the RGC axon, which has high energy requirements due to lack of saltatory conduction, has varicosities rich in mitochondria [15]. This portion of the optic nerve, in contrast to the myelinated one, is also rich in both cytochrome *c* oxidase (Complex IV) and succinate dehydrogenase (Complex II)—these two enzyme complexes form part of the electron transport chain (ETC) and their function has been linked to many other neurodegenerative diseases such as AD and PD [16]. Furthermore, neurons rely on astrocytes to supply precursors of the TCA cycle intermediates and other metabolites via the astro/glial shuttle. Under conditions of increased neuronal activity, the astrocyte–neuron lactate shuttle model allows astrocytes to metabolise glucose through anaerobic glycolysis to pyruvate and then to lactate, which is secreted to the extracellular space to be taken up by the neuron for further oxidative degradation. This model is thought to provide the neuron with more ATP than the classical/traditional metabolic pathway whereby neurons utilise glucose to make their ATP [17]. Animal studies have demonstrated that localised loss of metabolic support from astrocytes at the optic nerve head results in damage to the RGCs. The high energy demand of these astrocytes is demonstrated by their giant mitochondria [18]. There is further evidence that oxidative phosphorylation (OXPHOS) is particularly important to RGC dendrites. This is indicated by the intensive oxygen consumption in the inner plexiform layer (IPL), where RGC dendrites are located, and the high mitochondrial content in this region [19].

Several studies indicate that mitochondrial function may underlie both susceptibility and resistance to developing glaucoma (Table 1). A recent study exploring mitochondrial function in peripheral blood mononuclear cells (PBMC) of glaucoma patients demonstrated reduced systemic mitochondrial function in NTG compared to HTG patients [20]. The ultimate goal of the ETC is the production of ATP and the studies mentioned in Table 1 paint a clear picture of the involvement of mitochondrial dysfunction in glaucoma, with many reporting reduced ATP production. However, the decline in ETC activity associated with a deficiency in ATP, cannot solely explain the large spectrum of pathology observed [13]. RGCs, like all neuronal cells, do not proliferate, making them vulnerable to reactive oxygen species (ROS) insult. Such insults have the potential to disrupt the whole neuronal network. The mitochondrial ETC has been recognised as the main site for ROS generation [21]. Exposure to ROS can lead to the accumulation of oxidative damage to cellular components such as proteins and DNA, thus significantly impairing normal cellular function. Increased oxidative stress has been reported in POAG in various human studies (Table 2) and has also been shown to induce RGC death in experimental studies [22,23]. Such increased oxidative damage has the potential to affect DNA molecules and therefore cause a hyperactivation of PARP enzymes which in turn may result in a depletion of NAD^+^ and ultimately ATP (Section 5.2.1). The ETC couples the redox transfer of electrons from NADH (the reduced form of nicotinamide adenine dinucleotide (NAD^+^)) to oxygen, with the conversion of the electron motive force energy into a proton gradient across the mitochondrial inner membrane. NADH is a reducing equivalent, as it donates an electron to the electron acceptor in Complex I. Pathology could therefore arise from an excess of reducing equivalents (in this case NADH), known as reductive stress or pseudohypoxia, that ultimately results in the stalling of NAD^+^–coupled reactions, or a reduced proton gradient which in turn will impair pH and voltage–coupled processes, such as ATP synthesis by ATP synthase [24]. Reduced NAD^+^ levels cause a pseudo–hypoxia driven imbalance between nuclear and mitochondrial encoded OXPHOS subunits, which can be reversed by increasing NAD^+^ levels [25]. Depletion of NAD^+^ has been implicated in ageing and several neurodegenerative diseases. 

## 3. NAD^+^, NADH and Their Biological Functions

Nicotinamide adenine dinucleotide (NAD^+^) is an important coenzyme, central to several cellular bioenergetic functions. It acts as the parent molecule for the pyridine family of nucleotides (NADH, NADP and NADPH). NAD^+^ depletion will inevitably impair mitochondrial respiration and ATP synthesis resulting in energy crisis and cell death [52]. Over 300 dehydrogenases, including those located on the inner mitochondrial membrane which catalyse the transfer of electrons from NADH to coenzyme Q during OXPHOS, depend on NAD^+^ and NADH [53]. Furthermore, a number of important enzymes, such as poly (ADPribose) polymerase (PARP) and the sirtuin family of de–acetylase enzymes, rely on NAD^+^ as their primary substrate. PARP is a nuclear enzyme, which maintains genomic integrity by its involvement in DNA repair. It is activated by DNA strand breaks and it consumes NAD^+^ to make ADP ribose polymers. Chronic oxidative damage results in increased DNA damage and ultimately in NAD^+^ depletion, resulting in reduced ATP production and cell death [54]. NAD^+^ is also a substrate for NAD^+^ dependent de–acetylase enzymes, sirtuins (SIRT1–7). The central role of NAD^+^ in various cellular systems makes it an essential coenzyme for the health of the cell.

In mammals, NAD is made de novo from tryptophan, via the Preiss–Handler pathway from nicotinic acid (NA), via the salvage pathway from nicotinamide (NAM, the redox–active ring alone, without ADP–ribose), or via the nicotinamide ribose kinase pathway from nicotinamide riboside (NR) [55] (Figure 1). It is generally accepted that stable cellular NAD^+^ levels are maintained principally through the nicotinamide salvage pathway [56]. More importantly, in neurons, the major NAD^+^ precursor has been found to be nicotinamide [57]. Furthermore, a recent study reports of a trans–kingdom cooperation between bacteria and mammalian cells wherein bacteria contribute to the host’s NAD^+^ biosynthesis [58]. According to this study, the observed trans–kingdom interaction contributes substantially to the NAD^+^ boosting effect of oral NAM and NR supplementation, which in the future could be exploited as a means of therapeutically targeting NAD^+^ metabolism through manipulating the microbiome.

## 4. The NAD^+^/NADH Redox State

NAD^+^ and NADH can be either free or bound to protein, with three major pools compartmentalised in the nucleus, cytosol, and mitochondria [59]. The NAD^+^/NADH ratio is a regulator of cellular energy metabolism; glycolysis and OXPHOS. It reflects the metabolic status and redox state of the cell, and it fluctuates in response to a change in metabolism [60]. The NAD(H)–redox state is determined by the rate at which NADH is produced and the rate at which NADH is oxidised back into NAD^+^. NAD^+^ is reduced to NADH mostly in catabolic reactions such as glycolysis and the tricarboxylic acid (TCA/Krebs) cycle (Figure 2) [61]. To maintain a stable redox state, NADH needs to be re–oxidised to NAD^+^ constantly via several pathways. The oxidised form of NAD, NAD^+^, is the major form. The total intracellular NAD(H) concentrations (free and bound) have been reported to be 1–3 mM [62], with an [NAD^+^]total/[NADH]total ratio of 2–10/1 (depending on species, cell type, and metabolic state) [60,62,63,64]. However, the ratio in neuronal cell lines has been reported to be between 10 and 30 [65], indicating that neurons might have significantly higher NAD^+^/NADH ratio than non–neuronal cells. These values suggest that NAD^+^ can function as a metabolic regulator of the NAD^+^/NADH ratio in a variety of tissues. However, Zhang et al. demonstrated that the ratio of the free pool of NAD^+^ to NADH is closer to 600 (measured by the pyruvate/lactate ratio) in Cos–7 cell lines [66]. This highlights the importance of establishing which reported NAD^+^/NADH number represents the real situation in the cell type of interest. Although the NAD^+^/NADH redox ratio has been extensively studied in other organs, a description of the NAD^+^/NADH redox ratio in RGCs and its involvement in in POAG, remains to be elucidated. It is likely to be very cell– and context–dependent. It is important to investigate the possible causes for a reduced NAD^+^/NADH ratio. In theory, a reduced ratio can result from either a depletion in NAD^+^ or an increase in NADH or both (see Section 5 and Section 6). The total pool size of NAD^+^ and NADH, is, therefore, determined by the relative rates of their biosynthesis and degradation.

## 5. NAD^+^ Depletion

NAD^+^ declines in an age–related manner in multiple tissues [67] and can induce mitochondrial dysfunction and nuclear DNA damage [68] which may exacerbate neurodegenerative conditions. Multiple lines of inquiry have evaluated various NAD^+^ augmentation strategies, including treatment with the NAD^+^ precursors nicotinamide (NAM), nicotinamide riboside (NR), or nicotinamide mononucleotide (NMN) in conditions such AD, PD, and glaucoma as well as other axon degenerative injuries. These strategies have proved successful in several AD [69,70,71] and PD animal models [72,73,74]. The strong link between NAD^+^ and mitochondrial function suggests that bolstering cellular NAD^+^ levels can improve adaptive cellular stress responses in neurons. A study on 34 primary open–angle glaucoma (POAG) patients found lower plasma NAM levels compared to controls, suggesting that NAM supplementation might become a future therapeutic strategy [75]. Another study using a mouse model of an inherited glaucoma (DBA/2J; D2) found retinal levels of NAD decline with age, rendering RGC mitochondria vulnerable to stress factors. NAM is the major NAD^+^ precursor in neurons. High dose NAM supplementation in this animal glaucoma model was associated with an increase in retina NAD^+^ levels. At the highest doses, 93% of eyes had no detectable glaucoma [41]. A recent small randomised trial of 57 glaucoma patients, demonstrated that oral NAM (1.5–3 g/d) for 3 months significantly improved retinal function (measured by the photopic negative response indicating RGC health/activity) in 23% of glaucoma patients, compared to 9% of those on placebo [76]. A similar positive effect was seen on the visual fields. NAM was well tolerated and safe, suggesting its utility in the clinic, although further trials are required to assess its long–term neuroprotective effect in humans.

NAD^+^ depletion could be caused by an increase in NAD–consuming enzymes and/or a decrease in its synthesising enzymes. Some of the NAD–degrading enzymes in mammalian tissues include SIRTS, PARPs, CD38/CD157, and SARM1. These all compete with each other to consume cellular NAD^+^. Thus, the hyperactivation of one enzyme can impair the activities of other NAD^+^–dependent enzymes. These pathways have been shown to be mechanistically linked to axon degeneration and neurodegenerative diseases [7,41,74,77]. On the other hand, various enzymes involved in NAD^+^ synthesis, such as NMNAT2, have been demonstrated to be essential axon protection factors [77].

### 5.1. Synthesising Enzymes

NAD^+^ is synthesised via four different pathways (see Section 3), mediated by a number of enzymes (QPRT, NADSYN1, NAPRT, NAMPT, NMRK, NMNAT). De novo NAD^+^ synthesis from tryptophan is more prevalent in the kidney and considerably more in the liver. Other tissues, in contrast, rely almost exclusively on circulating NAM made by the liver, making the salvage pathway the predominant pathway for NAD^+^ biosynthesis [78]. Studies in neurons show that the major NAD^+^ precursor is NAM, making the salvage pathway the main one for NAD^+^ synthesis [57]. Exogenous NAM supplementation has been demonstrated to have very strong axonal protective effects [79] and is considered to be protective for neuron viability and brain function [80]. NAM is water soluble and can be administered orally. It therefore has potential as a safe, well–tolerated, and cost–effective agent [81] to be used in prospective studies on the clinical benefit of NAM supplementation in the treatment and management of glaucoma.

#### 5.1.1. NAM Salvage Pathway

Two classes of enzymes are essential to NAD^+^ synthesis via the NAM salvage pathway: nicotinamide phosphoribosyl transferase (NAMPT) and nicotinamide mononucleotide adenylyl transferases (NMNATs). NAMPT catalyses the production of nicotinamide mononucleotide (NMN) from NAM. NMNATs catalyse the formation of NAD^+^ from NMN. NMNATs are also involved in the de novo and Preiss–Handler pathways, in which they catalyse the formation of nicotinic acid adenine dinucleotide (NAAD) from NAMN [82].

Overexpression of NAMPT has been shown to increase cellular NAD^+^ contents and increase mitochondrial NAD^+^ levels along with resistance to apoptosis. Knockdown of NAMPT, on the other hand, produced opposite effects [83]. NMNATs are expressed as three protein isoforms (NMNAT1, NMNAT2, and NMNAT3) with differing cellular localisations. NMNAT1 is predominantly nuclear, NMNAT3 is mitochondrial and NMNAT2 is cytoplasmic; NMNAT2 is enriched in neurons and hair follicles [84]. Considering that neurons are primarily long tubes filled with cytoplasm, having a cytoplasmic NAD–producing enzyme in the axon is likely important. NMNATs, as NAD^+^ synthases, have critical roles in energy metabolism. Various studies show that each of these NAD^+^–synthesising enzymes play a critical role in the maintenance of neuronal survival and health and impact axon survival both under normal conditions and following injury [77,85,86,87,88]. Furthermore, within ophthalmology, NMNAT1 mutations (LCA9 gene) have been linked to retinal degeneration in Leber’s congenital amaurosis in humans [89]. NMNAT1 is essential for cell survival, and phenotypic manifestations of pathologic mutations in NMNAT1 present, amongst many other features, with optic nerve pallor [90]. Another example is NMNAT1–associated retinal degeneration, a recessive disease that causes severe vision loss during the first or second decade of life. Retinae of *Nmnat1^V9M/V9M^* mice supplemented with a normal copy of human *NMNAT1* via AAV–mediated gene augmentation showed preservation of structure and function for at least 9 months [91].

Failure of NMNAT2 to be transported to the axon results in the initiation of the degenerative program, thereby making NMNAT2 an essential axon protection factor [77]. Mutations that extend the half–life of NMNAT2 increase its ability to delay axon degeneration beyond that observed with the *Wld^S^* allele [92]. Furthermore, increasing stress negatively impacts NMNAT2 expression, indicating that a decline in NMNAT2 may induce vulnerability to axon degeneration in glaucoma [41]. Age–dependent decline in the expression of both NMNAT2 and NAMPT has been observed in retinal ganglion cells. Decline in NMNAT2 expression, which may occur in both an age– and IOP– dependent manner, results in depletion of NAD^+^ levels. This depletion has been prevented by nicotinamide treatment [41,93].

Pathological features of neurodegenerative diseases include neuron loss and axonal degeneration. In such conditions, including glaucoma, axon degeneration occurs before the neuronal cell body undergoes apoptosis [94]. Axon degeneration can also be induced directly by nerve injury in a process known as Wallerian degeneration. Discovery of the slow Wallerian degeneration allele (*Wld^S^*) in mice, in which severed axons could survive for long periods of time (weeks versus two days) [95], suggested that axon degeneration was not a passive process, but rather an active process initiated within axons. It also made it possible to confirm that axon loss in glaucoma models is mechanistically related to Wallerian degeneration [96,97,98]. *Wld^S^* protects axons after injury and in a number of models that fit the classical definition of Wallerian–like degeneration [99]. The *Wld^S^* gene results from the fusion of two genes; the NAD^+^ biosynthetic *Nmnat1* and *Ube4b* (an E4 ubiquitin ligase) genes [88,100]. The protein that this gene encodes, termed Wld^S^ is composed of the N–terminal 70 amino acids of Ube4b (termed N70), a unique 18–amino acid domain generated during the gene fusion event (termed Wld18), and the full–length sequence of NMNAT1. *Wld^S^* could therefore protect the axon through *N70*, *Wld18*, *NMNAT1*, or a combination of these domains. Avery et al. determined that by fusing the N–terminal 16 amino acids of *N70* (termed *N16*), together with *NMNAT1* (*N16–NMNAT1*) was sufficient to provide levels of axon protection equivalent to those observed with Wld^S^, in a fly model [101]. This indicates that Wld^S^–mediated axon protection results from NMNAT1 enzymatic activity and N16–dependent protein–protein interactions. NMNAT1 is located in the nucleus and is not essential for robust axon protection [88]. However, the Wld^S^ protein blocks axon degeneration by relocalising NMNAT1 into axons (thereby substituting for the loss of axon NMNAT2) [80,88,102,103]. N16 binds to the valosin–containing protein (VCP) which is essential for Wld^S^–like levels of axon protection [102]. In their study, Avery et al. observed that N16–VCP interactions may function to relocalise Nmnat1 outside the nucleus, perhaps to the cytoplasm or mitochondria, where it can exert its neuroprotective effects [101]. Severed axons exhibit a precipitous depletion of NAD^+^ and ATP just prior to fragmentation [79], which suggests that depletion of NAD^+^ might activate fragmentation. Overexpression of Nmnat1 in RGCs of D2 mice, through viral gene therapy, prevented glaucomatous nerve damage in >70% of treated eyes [41] suggesting that Nmnat1 alone was sufficient for neuroprotection. NMNATs therefore have a critical role in axon survival after injury, thereby indicating that the product of NMNAT activity, NAD^+^, is the key to axon protection. Furthermore, Kitaoka et al. evaluated the protective effect of NMNAT3 overexpression on optic nerve axonal protection in two different mouse models of glaucoma (the TNF injection model and the hypertensive glaucoma model). Overexpression of NMNAT3 exerted axonal protection against both TNF–induced and IOP elevation–induced optic nerve degeneration. Further, it was reported that the overexpression of NMNAT3 can alter the autophagy machinery, and NMNAT3 may be involved in decreased p62 and increased LC3–II levels in optic nerve degeneration [103].

#### 5.1.2. De Novo Pathway and QPRT

De novo NAD^+^ synthesis originates with tryptophan, and, through the kynurenine pathway (KP), results in NAD^+^ synthesis through quinolinic acid (QA) via the action of quinolinic acid phosphoribosyltransferase (QPRT)—Figure 3. QPRT catalyses the formation of nicotinic acid mononucleotide (NAMN) from QA, which is subsequently converted to nicotinic acid adenine dinucleotide (NAAD) by one of the three NMNAT enzymes. The final step is the amidation of NAAD by NAD synthetase (NADS), leading to the production of NAD^+^ [82].

QPRT is found primarily in glial cells and sporadically in neurons in the brain, indicating that tryptophan is not the major NAD^+^ precursor in neurons [104]. As mentioned above, organisms primarily use the NAM salvage pathway to generate NAD^+^. However, in a study looking at human monocyte–derived macrophages, blockade of the salvage pathway resulted in over 90% of NAD^+^ synthesis coming from the kynurenine pathway. In addition to this, it was also noted that QPRT expression decreases in aged macrophages. This decline in QPRT expression was associated with an induction of upstream KP metabolites culminating in an accumulation of QA but decreased levels of NaMN, NaAD, and NAD^+^ [105]. This ultimately results in supressed mitochondrial respiration. QPRT overexpression on the other hand was shown to prevent the accumulation of QA, restored NAD^+^ levels and increased oxidative phosphorylation, ECAR, and Complex I and II activities [105].

QA is considered to be involved in the pathogenesis of a number of inflammatory neurological diseases and there is now evidence for the KP being associated with Alzheimer’s disease [106]. Braidy et al. (2009) observed that in cultured human neurons and astrocytes, treatment with QA resulted in a dose–dependent increase in the activity of inducible and neuronal nitric oxide synthase (iNOS and nNOS, respectively). This led to an increase in cellular toxicity, NAD^+^ depletion and activation of PARP1. Inhibition of iNOS and nNOS on the other hand, was sufficient to rescue all of these effects, indicating that nitric oxide production likely plays a causative role in QA excitotoxicity [21]. This study found that QA acts as a substrate for NAD^+^ synthesis at very low concentrations (<50 nM) in both neurons and astrocytes but is cytotoxic at concentrations >150 nM in both cell types.

Astrocytes are the major cell type in the optic nerve head and are vital for retinal ganglion cell health. At each level in the optic nerve head, astrocytes are organised to support axons in their passage from the eye to the extraocular optic nerve. Since astrocytes are metabolically very active, they are vulnerable to physiological perturbations and are often the first cells in any neuronal system to respond to injury [107]. During the early stages of activation, astrocytes may act not only through a compromise in their supportive functions but also by a direct toxic effect on the retinal ganglion cell axons. Optic nerve head astrocytes contain nitric oxide synthase (NOS) the enzyme responsible for the production of nitric oxide (NO). NOS activity has been found to be upregulated in human and experimental glaucoma [108]. Excessive levels of NO will predispose to retinal ganglion cell death and exacerbate any disruption of gap junction–mediated intercellular communication in the astrocytes [109].

High concentrations of QA may therefore be associated with increased toxicity due to salvage pathway blockage, and a consequent shift into the KP pathway. This is turn may result in an overload of the system, which is not met by QPRT, resulting in reduced NAD^+^ and increased QA which is toxic to neurons and astrocytes. This is in addition to a deficit in NAD^+^, leading to mitochondrial dysfunction due to deficient NAD for mitochondrial energy production.

Furthermore, activation of the major metabolic pathway for tryptophan metabolism, the kynurenine pathway (KP), or a shift in the balance between the various branches of the KP, metabolising kynurenine, has been elucidated as one of the possible mechanisms involved in glaucomatous neurodegeneration [110]. As already mentioned on this section, some of the KP metabolites are known to display neurotoxic properties (e.g., quinolinic acid). However, other metabolites (such as kynureninase—KYNA), have been found to have neuroprotective properties [111]. In fact, retinal levels of KYNA have been shown to be elevated in response to RGC damage [112].

It is possible that blockage of the salvage pathway resulting in over 90% of NAD^+^ synthesis coming from the kynurenine pathway, as mentioned earlier in the sections, may cause a shift in the balance between the branches of the kynurenine pathway, resulting in an increase in QA and reduction in KYNA—the former being neurotoxic and the latter neuroprotective, thereby exposing RGCs to further stress.

### 5.2. Consuming Enzymes

#### 5.2.1. PARP

The poly (ADP–ribose) polymerases (PARPs) are a family of at least 18 enzymes involved in the maintenance of genomic stability and depend on NAD^+^ as substrate for their enzymatic function. They cleave NAD^+^ to release ADP–ribose (ADPR) groups that are used for the covalent mono– or poly(ADP–ribosyl)ation of proteins, DNA and RNA [113]. The majority (>90%) of PARylation is executed by PARP1, which participates in a number of necessary cellular processes, such as DNA repair, DNA/RNA metabolism, and cellular stress response. PARP–2 is the closest homolog to PARP–1 [7]. Activated PARP–1 under oxidative stress consumes NAD^+^ and depletes cellular ATP, eventually leading to energetic insufficiency and collapse [52]. PARP–1 activation results in the translocation of apoptosis–inducing factor (AIF) from mitochondria to the nucleus, fragmenting DNA [114]. PARP is activated by DNA strand breaks which occur as a result of DNA damage mainly mediated by ROS and nitric oxide [115]. During intense oxidative DNA damage, hyperactivation of PARP1, results in NAD^+^ depletion which then leads to PARP–1–mediated necrotic death of cells, parthanatos, which has been implicated in various age–related neurodegenerative diseases and accelerated ageing [114,116,117,118]. Increased oxidative stress has been reported in POAG and has been demonstrated to induce RGC death (Table 2) [22,23]. Such increased oxidative damage has the potential to affect DNA molecules and consequent hyperactivation of PARP enzymes. The overactivation of PARPs, such as in a state of increased oxidative damage, will cause a depletion of the total cellular NAD^+^ levels and thereby decrease its availability for other essential cellular processes. In fact, PARP–1 levels, measured in the aqueous humour of 41 POAG patients, were found to be higher when compared to 50 controls [119].

Inhibition of PARPs on the other hand, has been shown to increase NAD^+^ levels and delay neuronal death associated with mitochondrial dysfunction in a PD fly model [74]. NAM is one such inhibitor of PARP–1 [120]. NAM supplementation has been shown to be protective against retinal ganglion cell neurodegeneration and prevented a number of early gene expression changes in D2 retinal ganglion cells [41].

#### 5.2.2. CD38/CD157

CD38 is one of the main NAD–degrading enzymes in mammalian tissues [121,122]. It catalyses the synthesis of the Ca^2+^responsive messenger cyclic ADP–ribose (cADPR) by use of NAD^+^ and plays a key role in multiple physiological processes such as immunity, metabolism, inflammation, and even social behaviours [123]. Despite CD38 being a lymphocyte differentiation antigen, it is also expressed in neurons [124], astrocytes [125], and microglial cells [126]. There is an age–dependent increase of CD38 [127], which may contribute to cellular NAD^+^ depletion and impaired mitochondrial function observed in neurodegenerative diseases of ageing, such as PD, AD and glaucoma. Cells overexpressing CD38 are more susceptible to oxidative stress, as they have lower NAD^+^ levels and a reduction in proteins associated with antioxidant defence [128]. CD38 is also implicated in the degradation of NMN [127] thereby indicating that an increase in activity could not only degrade NAD^+^ but also reduce its synthesis.

CD38 has been shown to control NAD bioavailability and the activity of NAD–dependent enzymes [129]. Braidy et al. found that silencing CD38 expression using siRNA in rat cortical neuron cultures increased NAD levels by 5–fold through its NADase activity. Experimental data using CD38 knock–out mice has demonstrated positive effects of CD38 deletion against neurodegeneration and neuroinflammation [130]. CD157, just like CD38, is also a member of the ADP–ribosyl cyclase family of enzymes that catalyse the formation of NAM, generation cADPR and ADPR from NAD^+^. However, its efficiency in generating cADPR is lower than that of CD38 [131]. Nevertheless, the *BST–1/CD157* gene (which codes for CD157) has recently been associated with Parkinson’s disease [132].

#### 5.2.3. SARM1

Sterile alpha and Toll/interleukin–1 receptor motif–containing 1 (SARM1) is a member of the Toll/IL–1 Receptor (TIR) domain–containing superfamily [133]. It is a newly recognised class of NADase that cleaves NAD^+^ into NAM, ADPR, and cADPR via its TIR domain to trigger axon destruction. Axonal injury induces NAD^+^ loss [79]; SARM1 is required for this injury–induced NAD^+^ depletion both in vitro and in vivo [85,134]. Progressive axon degeneration defines multiple neurodegenerative diseases. As mentioned in Section 5.1.1, these diseases are often termed Wallerian–like, because of the similar neuronal death morphology and mechanism to Wallerian degeneration [96,97,98]. Mitochondria play a role both in the late stages of Wallerian degeneration after axon transection [135], but also at an early step of the Wallerian pathway, upstream of NMNAT2 [136]. An upregulation of SARM1 activity, triggers a rapid collapse of NAD^+^ levels and increases neuronal degeneration [133]. Activation of SARM1 is sufficient to deplete NAD^+^ levels and initiate the Wallerian degeneration pathway [134]. On the other hand, SARM1 deficiency has been shown to protect against axonal degeneration in several models of neurodegenerative conditions [137,138,139,140]. Overexpression of NMNATs and SARM1 deletion in sensory neurons delays axon degeneration caused by rotenone (mitochondrial complex–1 inhibitor), thereby showing that mitochondrial dysfunction induces SARM1–dependent cell death [141]. In RGCs, kainic acid (KA)–mediated upregulation of SARM1 has been shown to promote Wallerian–like degeneration [142]. Furthermore, studies in mice have shown that *SARM1* deletion is as effective as *Wld^S^* in preventing axon degeneration [143,144]. These findings indicate that SARM1 and Wld^S^ participate in the same RGC axon degeneration pathway, with SARM1–mediated NAD^+^ depletion contributing to axon degeneration [85] and *Wld*^S^ expression compensating for NAD^+^ depletion after axon injury [79,87]. SARM1 has also been found to be necessary for RGC axon loss and cell death, as well as oligodendrocyte loss in the optic nerve, in a neuroinflammatory D2 mouse model of glaucoma induced by intravitreal TNF–α injection [96]. ATP levels in the optic nerve of the D2 mouse dramatically reduce at 6 months, before any loss of optic nerve occurs. In a different study, using the mouse optic nerve crush (ONC) model, SARM1 deficiency was able to protect axons but not the soma from degeneration resulting ultimately in RGC death [143]. As mentioned before, SARM1 is an NAD^+^ cleaving enzyme, which when activated induces axonal NAD^+^ loss. Such a loss in NAD^+^ levels will impact both glycolysis and OXPHOS, thereby resulting in reduced ATP levels, indicating that NAD^+^ supplementation may have protective effects in optic nerve preservation [41].

#### 5.2.4. Sirtuins

Sirtuins are a class of proteins that catalyse deacetylation and ADP ribosylation, thereby modifying a great number of proteins, including histone and non–histone proteins. They require NAD^+^ for their activity, with each cycle of the Sirtuin catalysed reaction, consuming one equivalent of NAD^+^ and thereby likely influencing the NAD^+^: NADH ratio in the cells [145]. However, Sirtuin activity can be differentially regulated by the cellular concentrations of both NAD^+^ and NAM, as such the intracellular NAD^+^/NAM ratio may be a better predictor of sirtuin activity than the NAD^+^/NADH ratio [146].

In mammals, there are seven sirtuin enzymes (SIRT1–SIRT7). Three sirtuins are located in the mitochondria (SIRT3–SIRT5), while SIRT1, SIRT6 and SIRT7 are predominantly located in the nucleus, and SIRT2 is found in the cytoplasm (Table 3). Sirtuins use NAD^+^ as a cosubstrate to remove acetyl moieties from lysines on histones and proteins, releasing NAM and O–acetyl ADP–ribose. They have multiple functions from regulating DNA damage, mitochondrial biogenesis, ATP production, cell signalling, DNA repair and many more (Table 3).

As mentioned in the previous sections, RGC loss in glaucoma models is mechanistically related to Wallerian–like degeneration [96,97,98]. It is considered that SIRT1 contributes to preservation of neurons from Wallerian degeneration, whilst this neuroprotective effect is blocked by the SIRT1 inhibitor sirtinol and by SIRT1 silencing with siRNA [78,105,166]. Resveratrol treatment and SIRT1 overexpression, for instance, have been shown to delay RGC loss and reduce oxidative stress following optic nerve crush [166]. It is worth mentioning here that, whilst the other NAD^+^ consuming enzymes mentioned above were damaging to RGCs, SIRTs on the other hand are protective. However, in the event when there is a depletion in NAD^+^ levels, the protective activity of SIRTs will be reduced, thereby placing the RGCs at higher risk of damage.

SIRT1 is expressed throughout the retina, including the retinal ganglion cell layer, inner retinal layer cells [167], photoreceptor cells, and retinal pigment epithelium [168]. Studies have indicated that downregulation of SIRT1 is involved in ocular ageing and retinal neuron degeneration, whilst its upregulation in neuroprotection [169,170]. It may therefore protect against optic nerve degeneration in glaucoma patients. SIRT3 is present in the mitochondrial matrix and can regulate mitochondrial function [153]. Considering observed mitochondrial dysfunction in glaucoma, SIRT3 activation may be a line of work to promote improved mitochondrial function. Glaucomatous human retinae have shown a 2–fold increased expression of SIRT3 compared to normal retinae. In addition, human glaucomatous retinae have shown increased expression of SIRT1, SIRT3, SIRT6, and SIRT7 compared to age–matched non–glaucomatous controls [171]. Changes in NAD^+^ concentrations have been linked to corresponding changes in Sirtuin activity. Considering Sirtuins are expressed in the RGC layer of the retina and have protective effects on RGCs [172], their dysfunction may be closely related to the pathogenesis of glaucoma, making Sirtuins a new potential target for glaucoma treatment.

## 6. Increased NADH

NADH is mostly produced in the cytosol by glycolysis and in the mitochondria by the tricarboxylic acid (TCA) cycle (Figure 2). Mitochondrial NADH is oxidised to NAD^+^ at mitochondrial respiratory complex I (NADH dehydrogenase) of the ETC. During ATP synthesis, levels of NAD^+^ and NADH are tightly regulated within a cell, whereby an excess of NADH leads to increased reductive stress and ultimately ROS production [173]. Complex–I impairment may therefore lead to an increase in the NADH levels and a decrease in the NAD^+^/NADH ratio, leading to a reduced state within the mitochondrial matrix and ultimately reduced ATP levels. Defects in Complex–I–related OXPHOS function have been associated with a wide spectrum of neurodegenerative diseases, including glaucoma/NTG [32,33]. Furthermore, an excess of NADH inhibits the enzymes that reduce NAD^+^ to NADH (such as glyceraldehyde 3–phosphate dehydrogenase and dihydrolipoamide dehydrogenase in the pyruvate dehydrogenase complex) resulting in an increase in reactive oxygen species (ROS) production [174,175].

Van Bergen et al. demonstrated an increase in total NADH levels in Leber’s hereditary optic neuropathy (LHON) lymphoblasts versus age–matched controls [32]. The higher total NADH in LHON lymphoblasts also corresponded with a significant decrease in the NAD^+^/NADH ratio compared to age–matched controls. However, whilst both POAG and LHON had reduced Complex–I activity, total NADH levels and the NAD^+^/NADH ratio remained unchanged in POAG lymphoblasts versus age–matched controls [32]. The degree of heteroplasmy may explain this finding, as LHON lymphoblasts have a more severe mitochondrial defect (Complex–I), compared to POAG. A certain amount of defective Complex–I must be present before oxidative dysfunction occurs and clinical signs become apparent; this is known as the threshold effect [176]. The threshold for disease is lower in cells/tissues that are highly dependent on oxidative metabolism, such as RGCs versus tissues that are not, such as lymphocytes/lymphoblasts. RGCs will therefore be especially vulnerable to the effects of pathogenic mutations causing defects in the ETC Complexes. As such, it may be that whilst POAG lymphoblasts may be unaffected with regard to NADH level, RGCs may exhibit high NADH levels and reduced NAD^+^/NADH ratio, considering the high energy demand these cells have compared to lymphoblasts. Another explanation for this finding may also relate to the fact that the cell model in this study (lymphoblasts) were grown in vitro and removed from their in vivo context. As a result, various parameters of the redox state of the cell and their gene expression will differ and therefore may not necessarily represent the cells as they would be in situ in the human body [177]. It is important to note that such defects in Complex–I noted by the studies presented in this section, are systemic and not solely induced in RGCs as a consequence of high IOP. This has two major implications (i) it opens up the possibility of mitochondrial function related metabolites as systemic biomarkers for glaucoma diagnosis, progression and management, and (ii) it may lead to further understanding of the pathogenesis of glaucoma when IOP is within the statistically normal range (NTG), or its progression when IOP is well controlled through glaucoma treatment (e.g., eye drops, surgery).

## 7. Discussion

The pathophysiology of glaucoma is complex, with multiple possible mechanisms that may lead to RGC degeneration. Besides the main modifiable risk factor of raised IOP, various studies have shown that other risk factors are involved in the onset of the condition, as demonstrated clearly in NTG. At present the main treatment for managing glaucoma is by reducing IOP through various methods such as medications, laser or surgery; however, patients may still deteriorate despite IOP lowering. In NTG patients develop glaucoma with statistically normal IOP levels (less than 21 mmHg), while most patients with ocular hypertension (OHT) do not develop glaucoma [178], suggesting that IOP is only one of several important risk factors in glaucoma pathogenesis.

The diagnosis and assessment of glaucoma progression requires a detailed examination assessing both the structure (optic disc assessment) and function (visual field testing) of the optic nerve. Unfortunately, glaucoma screening has an estimated specificity of approximately 85% [179] thereby resulting in an insufficient predictive power. As such, several patients may have glaucoma before being diagnosed with it. In developed countries alone, at least half of all glaucoma patients remain undiagnosed [180]. This number goes up to 90% worldwide [181]. This is especially important, when it comes to early diagnosis, which is critical to managing glaucoma progression and preventing further irreversible sight loss. This underlines the strong demand for additional diagnostic options such as a biomarker for disease diagnosis, risk profiling and treatment monitoring. A biomarker is defined as an objectively measurable indicator in normal biological processes, pathogenic processes, or in response to a therapeutic intervention, and therefore has the potential to be used as an indicator of disease detection and monitoring [182].

The RGC axons have a high density of mitochondria, required to sustain their high energy demand from mitochondrial oxidative phosphorylation, thereby making these cells particularly sensitive to mitochondrial dysfunction. As shown throughout this review, mitochondrial dysfunction is associated with several diseases of the eye, including POAG, LHON and ADOA. LHON, for instance, is caused by mutations to Complex I genes and Complex I deficiency has been shown in POAG patients. It is important to stress here that the mutated proteins in all these diseases are present in all cells, not just RGCs, and can be observed in peripheral tissue, such as blood cells [177]. The answer to the question as to why such mutations result in apoptosis only of RGCs is unknown, but it is hypothesised to be due to the high energy demand of these cells. Analysing primary patient tissue for these diseases is limited to post–mortem biopsies. On the other hand, generating RGCs from human induced pluripotent stem cells (iPSC) may be beneficial for studying disease pathways in the target tissue involved. However, this is an expensive and laborious process, making such cells a suboptimal biomarker cell model. To date, little has been done on patient–derived iPSC RGCs [183] The question, therefore, is whether nontarget cells (lymphocytes, lymphoblasts, fibroblasts etc.) can be used to detect difference in the NAD^+^/NADH redox state in glaucoma. Lymphocytes for instance, can be obtained from blood, which is by far the most accessible tissue and samples can be obtained from the same participants on multiple occasions. Can current assays for detecting NAD^+^(H) levels detect changes between disease and control in such cell models? Van Bergen et al. argues that POAG lymphoblasts have a mitochondrial defect which is more modest than LHON lymphoblasts and not severe enough to alter the redox status of the lymphoblast cells [32]. However, several studies in human and animal models have shown that there is evidence that the NAD^+^/NADH redox state may be a useful biomarker for monitoring and management of several diseases [184,185,186,187,188,189,190,191]. Mitochondrial dysfunction in POAG has been reported by many studies in human lymphocyte, lymphoblast and fibroblast models (Table 1), raising the point that if NAD levels underlie such dysfunction, then one might expect to be able to measure such differences in these cells.

As described in the previous sections, mitochondria oxidise NADH to NAD^+^ and as such mitochondrial dysfunction will result in a decrease in the NAD^+^/NADH ratio. On the other hand, upregulation of NAD^+^ synthesising enzymes or downregulation of NAD^+^ consuming enzymes would result in altered NAD^+^ levels, and ultimately an altered NAD^+^/NADH ratio. Furthermore, the NAD^+^/NADH ratio would act as a crucial indicator of NAD^+^ levels within the cell, should potential strategies for NAD^+^ augmentation such as oral supplementation (e.g., NAD^+^ precursors such as NAM) or gene therapy (e.g., Section 5.1.1 NMNAT1, NMNAT2) be employed.

## 8. Conclusions

Throughout this review we documented evidence of mitochondrial dysfunction in glaucoma. Although the precise mechanism of this dysfunction is still unclear, studies in various neurodegenerative diseases such as AD, PD, and now in glaucoma have shown a reduction in NAD^+^ levels with or without a reduced NAD^+^/NADH ratio to be associated with mitochondrial dysfunction. Mitochondrial dysfunction observed in glaucoma could be the result of reduced NAD^+^ levels (e.g., as a consequence of an age–dependent NAD^+^ decline), it may be the cause of a reduction in NAD^+^ levels that in the presence of local factors, such as IOP, can lead to local RGC damage, or it may be brought about by an IOP–dependent NAD^+^ decline [192].

The reduction in systemic mitochondrial function observed in glaucoma gives hope for NAD^+^ and/or the NAD^+^/NADH ratio to be used as a biomarker, measured, for instance, in lymphocytes during a routine blood test. Metabolic dysfunction and mitochondrial abnormalities have been shown to occur prior to glaucomatous neurodegeneration [41], thereby making NAD^+^ levels and NAD^+^/NADH ratio potential biomarkers, facilitating glaucoma risk profiling in clinic. Furthermore, various NAD^+^ augmentation strategies in AD, PD and glaucoma animal models have proved successful in both disease prevention and delay of progression. Both NAD^+^ levels and the NAD^+^/NADH redox state may therefore prove useful clinical biomarkers following treatment with such augmentation strategies to support disease monitoring and response to treatment.

## Figures and Tables

**Figure 1 cells-10-01402-f001:**
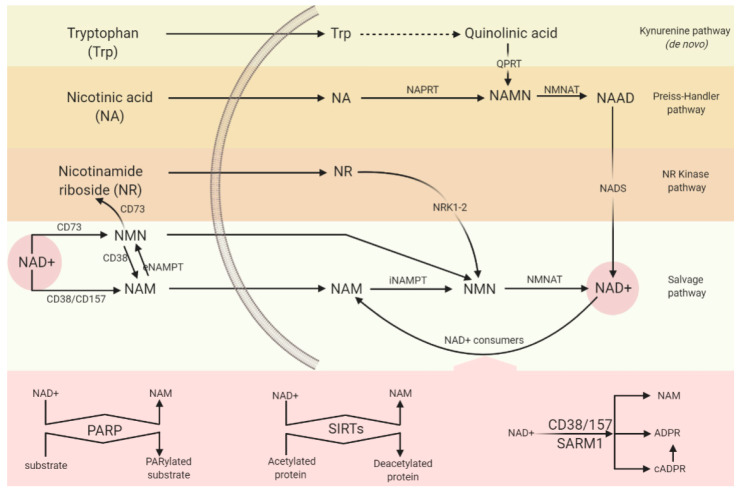
NAD^+^ synthesis and catabolism. There are multiple routes to sustain NAD concentrations within the cell; de novo from tryptophan (Trp), through the Preiss–Handler pathway, or via nicotinamide (NAM) or nicotinamide riboside (NR) salvage pathways. Extracellular is shown to the left of the image, intracellular to the right. NAD^+^ consumption is shown in pink. Legend: Nicotinamide mononucleotide (NMN); extracellular nicotinamide phosphoribosyltransferase (eNAMPT); intracellular nicotinamide phosphoribosyltransferase (iNAMPT); nicotinate phosphoribosyltransferase (NAPRT); nicotinic acid mononucleotide (NAMN); quinolinate phosphoribosyltransferase (QPRT); nicotinic acid adenine dinucleotide (NAAD); nicotinamide riboside kinase 1 and 2 (NRK1–2); NAD^+^ synthase (NADS); NA phosphoribosyl–transferase (NAPRT); NAM mononucleotide transferases (NMNAT); poly(ADP–ribose) polymerases (PARP); sirtuins (SIRTs); Sterile alpha and TIR motif–containing 1 (SARM1); cyclic ADP–ribose synthases (cADPR).

**Figure 2 cells-10-01402-f002:**
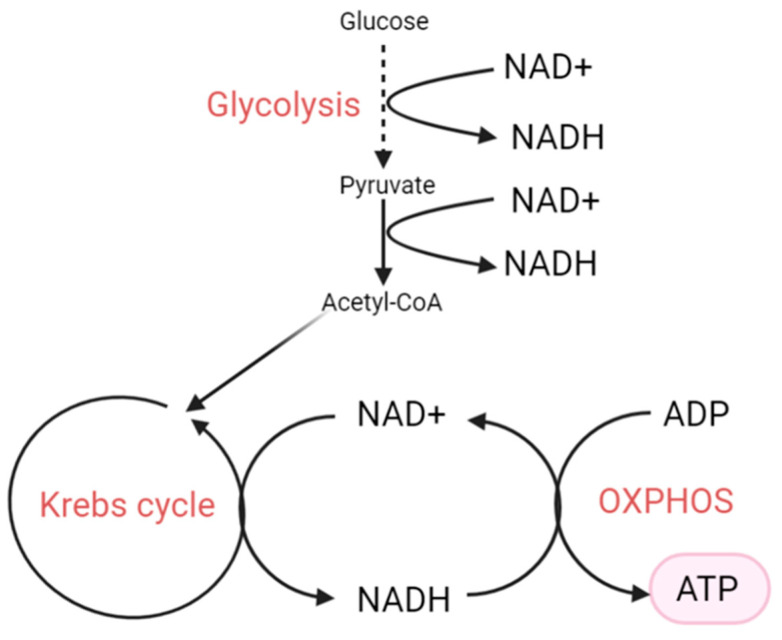
NAD^+^(H) redox involvement in ATP production.

**Figure 3 cells-10-01402-f003:**
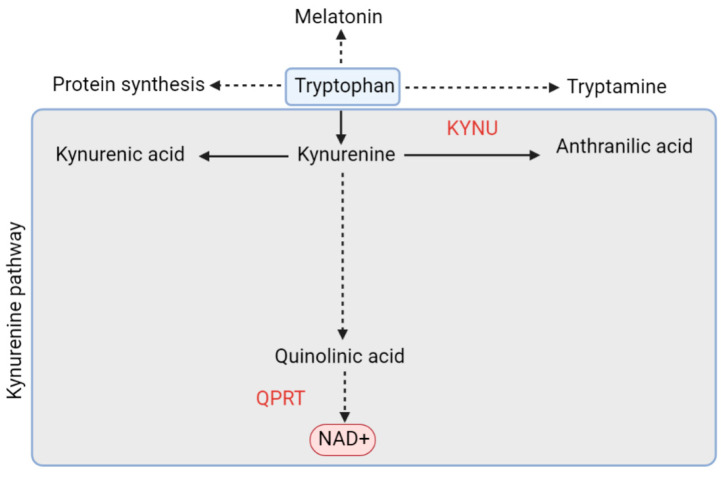
Simplified overview of the kynurenine pathway of tryptophan metabolism. Key enzymes are shown in red: kynureninase (KYNU); quinolinic acid phosphoribosyltransferase (QPRT).

**Table 1 cells-10-01402-t001:** Mitochondrial involvement in glaucoma patients and animal models of glaucoma.

Study	Finding
Ju et al. 2009 [26]	Elevated hydrostatic pressure triggered mitochondrial changes and altered OPA1 gene expression before the onset of apoptosis in differentiated RGC–5 cells
Aung et al. 2002; Powell et al. 2003; Yu–Wai–Man et al. 2010 [27,28,29,30]	Polymorphisms in the OPA1 gene are associated with NTG, and they also influence the phenotypic feature in patients with HTG.
Abu–Amero et al. 2006 [31]	Reduced mitochondrial respiratory activity in lymphocytes of POAG patients compared controls
N. J. Van Bergen et al. 2015a [32]	Reduced Complex–I enzyme specific activity and ATP synthesis in POAG lymphoblasts
[33]	Complex I defect in POAG lymphoblasts, leading to decreased rates of respiration and ATP production
Wolf et al. 2009 [34]	Association of NTG with common sequence variants of OPTN, MFN1, MFN2 and PARL
X. Hu et al. 2018 [35]	OPA1 overexpression may protect RGCs by ways of enhancing mitochondria fusion and parkin mediated mitophagy
Bailey et al. 2016; Khawaja et al. 2016; Khawaja et al. 2018; Sundaresan et al. 2015 [36,37,38,39]	Various genes encoding for mitochondrial proteins have been found to be associated with POAG, and in particular NTG, including TXNRD2, ME3, VPS13C, GCAT, PTCD2, ND5
Lascaratos et al. 2015 [40]	Resistance to developing glaucoma is associated with systemic mitochondrial efficiency
Williams et al. 2017 [41] Fraenkl et al. 2011; Goldblum et al. 2010 [42,43]	Metabolic dysfunction and mitochondrial abnormalities occur prior to glaucomatous neurodegenerationReduced plasma citrate levels in patients with glaucoma compared to controls—citrate is a major component in mitochondrial metabolism

**Table 2 cells-10-01402-t002:** Oxidative stress in POAG—human studies.

Study	Finding
Ferreira et al. 2004 [44]	Reduced levels of water–soluble antioxidants (glutathione, ascorbate, tyrosine) in aqueous humour of POAG compared to controls
Izzotti et al. 2003; Saccà et al. 2005 [45,46]	Oxidative DNA damage is exaggerated in the trabecular meshwork of POAG patients
Gherghel et al. 2005 [47]	Glaucoma patients have lower serum GSH and total glutathoine (t–GSH) levels as compared with age–matched controls
Yildirim et al. 2005 [48]	Malonyldialdehyde (marker of oxidative stress) levels were more than 2–fold greater in the serum of POAG patients as compared with healthy controls
Tanito et al. 2012 [49]	Biological antioxidant potential level, a measure of total antioxidative stress activity, was lower in plasma in the POAG and pseudo–exfoliation syndrome groups compared with the control groups
Sorkhabi et al. 2011 [50]	Increased oxidative DNA damage in the serum and aqueous humour of glaucoma patients
Yuki et al. 2010 [51]	Increased serum total antioxidant and decreased 8–hydroxy–2′–deoxyguanosine in response to increased systemic oxidative stress in patients with normal–tension glaucoma

**Table 3 cells-10-01402-t003:** Localisation and activity of sirtuins in mammals.

Sirtuin	Activity	Location	Biological Function	References
SIRT1	Deacetylation	Nucleus	Regulation of DNA damage, stress response, mitochondrial biogenesis, glucose and lipid metabolism	[147,148,149,150]
SIRT2	Deacetylation	Cytosol	Lipid and glucose metabolism, control of cell cycle	[151,152]
SIRT3	Deacetylation	Mitochondria	Regulation of ATP production, metabolism, apoptosis, cell signalling	[153,154,155]
SIRT4	ADP–ribosylation	Mitochondria	Inhibition of insulin secretion, repression of fatty acid oxidation, tumour suppressor	[156,157,158]
SIRT5	Deacetylation	Mitochondria, cytosol	Urea cycle, ATP production, glycolysis	[159,160,161]
SIRT6	Deacetylation, ADP–ribosylation	Nucleus	Genomic stability and repair, metabolism and aging	[162,163,164]
SIRT7	Deacetylation, ADP–ribosylation	Nucleolus	DNA repair, ageing	[165]

## Data Availability

Not applicable.

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
