# Peer review of "Neuroprotection in Glaucoma: NAD+/NADH Redox State as a Potential Biomarker and Therapeutic Target"

_cells, 2021, doi:10.3390/cells10061402_

Round 1

Reviewer 1 Report

In this manuscript, the authors have reviewed in detail the roles for NAD+/NADH in cellular, and specifically RGC, metabolism and the ways in which alterations in mitochondrial function and oxidative stress might play a role in RGC death in glaucoma. Further, they have discussed NAD-interacting enzymes and substrates as targets for RGC neuroprotection. Finally, they mention the idea of assaying NAD+ concentrations as a potential biomarker for glaucoma.

Overall this is a very nice review article, which is comprehensive and well written. I have only a few suggestions below, which may be of interest for the authors to consider. Most notably, I think it would help to expand upon what I view as the most interesting aspect of this review (and what is least available in other published works): the potential utility of assaying this pathway as a biomarker for glaucoma.

  • The authors note on page 17 that lymphoblasts from patients with LHON exhibit reduced levels of NAD+ versus controls, whereas those from POAG patients do not. They offer as a potential explanation that lymphoblasts may be less dependent on oxidative metabolism, and perhaps RGCs would be more vulnerable to effects on ETC complexes and manifest lower NAD+. I am intrigued by the potential utility for measuring NAD+ levels in patient cell samples as a potential biomarker for glaucoma – this seems to be a point emphasized by the authors as well given that, although this idea involves only a few paragraphs at the end of the manuscript, it is highlighted in the title and the abstract. To this end, I would recommend the background and ideas regarding biomarkers be expanded to some extent. I was interested to review reference 180 and find that the samples being tested were not peripheral blood leukocytes, but rather lymphocyte cell lines that were developed following transformation with EBV. Does this alter the oxidative demand of the cells? If the authors wanted to propose a more physiologically relevant assay for glaucoma, what do the authors think about developing iPS lines from peripheral blood monocytes of patients, and then differentiating them into RGCs to then measure NAD+ levels? Labor intensive, perhaps, but so is EBV-mediated line generation. Direct-to-neuron conversion protocols may also facilitate more efficient assays.

  • Please qualify the statement made on lines 75-77. “…patients with NTG and high-tension glaucoma (HTG) still go blind despite treatment [9]”. Most patients with glaucoma are not blind and will not go blind within their lifetime. Most glaucoma blindness is attributable to late diagnosis. Bilateral blindness is much less common than unilateral blindness. The scenario in which a diagnosed patient is under care and progresses to blindness despite treatment is the exception rather than the rule. Please see: https://pubmed.ncbi.nlm.nih.gov/12689894/ https://pubmed.ncbi.nlm.nih.gov/26302445/ While I am all for making the case that new treatments for glaucoma are urgently needed, it is important not to overstate the risk of blindness for patients already under care. Simply rephrasing to something like “an important subset of patients with NTG…still lose significant vision despite treatment” seems reasonable.

Minor points:

  • The point about giant mitochondria in optic nerve head astrocytes is redundantly made twice on lines 87 and 101.

  • Line 99: “neurons” is misspelled.

Author Response

In this manuscript, the authors have reviewed in detail the roles for NAD+/NADH in cellular, and specifically RGC, metabolism and the ways in which alterations in mitochondrial function and oxidative stress might play a role in RGC death in glaucoma. Further, they have discussed NAD-interacting enzymes and substrates as targets for RGC neuroprotection. Finally, they mention the idea of assaying NAD+ concentrations as a potential biomarker for glaucoma. Overall this is a very nice review article, which is comprehensive and well written. I have only a few suggestions below, which may be of interest for the authors to consider. Most notably, I think it would help to expand upon what I view as the most interesting aspect of this review (and what is least available in other published works): the potential utility of assaying this pathway as a biomarker for glaucoma.

The Authors would like to thank the Reviewer for their comments with which to improve the manuscript. 

The authors note on page 17 that lymphoblasts from patients with LHON exhibit reduced levels of NAD+ versus controls, whereas those from POAG patients do not. They offer as a potential explanation that lymphoblasts may be less dependent on oxidative metabolism, and perhaps RGCs would be more vulnerable to effects on ETC complexes and manifest lower NAD+. I am intrigued by the potential utility for measuring NAD+ levels in patient cell samples as a potential biomarker for glaucoma – this seems to be a point emphasized by the authors as well given that, although this idea involves only a few paragraphs at the end of the manuscript, it is highlighted in the title and the abstract. To this end, I would recommend the background and ideas regarding biomarkers be expanded to some extent.

The Authors agree that tying the theme of this manuscript together requires additional hypothesis generation towards the end of the manuscript. We have added the following to the text:

“The RGC axons have a high density of mitochondria, required to sustain their high energy demand from mitochondrial oxidative phosphorylation, thereby making these cells particularly sensitive to mitochondrial dysfunction. As shown throughout this review, mitochondrial dysfunction is associated with several diseases of the eye, including POAG, LHON and ADOA. LHON, for instance, is caused by mutations to complex I genes and complex I deficiency has been shown in POAG patients. It is important to stress here that the mutated proteins in all these diseases are present in all cells, not just RGCs, and can be observed in peripheral tissue, such as blood cells [188]. The answer to the question as to why such mutations result in apoptosis only of RGCs is unknown, but it is hypothesised to be due to the high energy demand of these cells. Analysing primary patient tissue for these diseases is limited to post-mortem biopsies. On the other hand, generating RGCs from human induced pluripotent stem cells (iPSC) may be beneficial for studying disease pathways in the target tissue involved. However, this is an expensive and laborious process, making such cells a suboptimal biomarker cell model. To date, little has been done on patient-derived iPSC RGCs [189]  The question, therefore, is whether non-target cells (lymphocytes, lymphoblasts, fibroblasts etc.) can be used to detect difference in the NAD+/NADH redox state in glaucoma. Lymphocytes for instance, can be obtained from blood, which is by far the most accessible tissue and samples can be obtained from the same participants on multiple occasions. Can current assays for detecting NAD+(H) levels detect changes between disease and control in such cell models? Van Bergen et al. argues that POAG lymphoblast have a mitochondrial defect which is more modest than LHON lymphoblasts and not severe enough to alter the redox status of the lymphoblast cells [32]. However, several studies in human and animal models have shown that there is evidence that the NAD+/NADH redox state may be a useful biomarker for monitoring and management of several diseases [190-197]. Mitochondrial dysfunction in POAG, has been reported by many studies in human lymphocyte, lymphoblast and fibroblast models (Table 1), raising the point that if NAD levels underlie such dysfunction, then one might expect to be able to measure such differences in these cells”

I was interested to review reference 180 and find that the samples being tested were not peripheral blood leukocytes, but rather lymphocyte cell lines that were developed following transformation with EBV. Does this alter the oxidative demand of the cells?

This reference is now number 32, and not 180- this happened automatically  whilst the paper was edited. 

The Reviewer raises an important point about whether transformed cells retain their parental phenotypes and whether genetic systemic, or even susceptibility to epigenetic changes and regulation, drive changes in these cells.  We have added the following to the text:

“Another explanation for this finding may also relate to the fact that the cell model in this study (lymphoblasts) were grown in vitro and removed from their in vivo context. As a result, various parameters of the redox state of the cell and their gene expression will differ and therefore may not necessarily represent the cells as they would be in situ in the human body [183]”

It is important to note that both VanBergen et al.  and this review presented here do not discuss NAD+ in LHON, but rather reduced Complex-1 activity in both LHON and POAG and increased NADH in LHON causing a reduction in NAD+/NADH ratio. The justification for this is that LHON lymphoblasts with a more severe mitochondrial defect (complex-I enzyme activity and growth under galactose) have imbalanced redox levels in this cellular model. The POAG lymphoblast mitochondrial defect is more modest than in LHON lymphoblasts and is not severe enough to alter the redox status of the lymphoblast cells.

This raises the question the Reviewer points out: Does the cell type affect the redox state of the cell?

The answer to this is complex and requires mechanistic studies in isolated cell types which are yet to be done in the context of a human ophthalmic disease. Nevertheless, it is reasonable to assume that different cells (as we already know) have different metabolic requirements. VanBergen and others have reported increased mtDNA copy numbers and also increased mitochondrial mass, presumably in response to the cells transforming from resting cells into actively proliferating cells. This higher mitochondrial content makes the cells more oxidative than other cell types, such as fibroblasts, and therefore, subtle differences in mitochondrial respiration may be detected more easily. However, as these cells are in vitro and removed from their in vivo context, parameters of the redox state of the cell, their gene expression will differ and maybe not necessarily represent the cells as they would be in situ in the human body.

It is important to mention that from our experience, as well as from other reported studies, measurement of NAD+ and NADH levels is difficult. In fact, a subsequent study by Lopez Sanchez et al. detected no difference in NAD+/NADH ratios in LHON Lymphoblasts compared to controls, so it is as yet unclear how redox potentials are changed, or whether this is cell and context dependent. How this relates to the myriad of glaucoma patients we see in the clinics is unknown and will require studies performed en masse, likely in the setting of a clinical trial.

Nevertheless, we and others have been able to identify differences in mitochondrial respiration in lymphocytes and lymphoblasts. If NAD levels underlie these differences, then one might expect to be able to measure NAD differences in these cells. These will be important experiments to perform in the future.

If the authors wanted to propose a more physiologically relevant assay for glaucoma, what do the authors think about developing iPS lines from peripheral blood monocytes of patients, and then differentiating them into RGCs to then measure NAD+ levels? Labor intensive, perhaps, but so is EBV-mediated line generation. Direct-to-neuron conversion protocols may also facilitate more efficient assays.

The Authors completely agree with these thoughts and this is an active area of research for many labs, including our own. However, to date, little has been done on patient-derived iPSC RGCs.

This is mentioned in line 554-557:

“On the other hand, generating RGCs from human induced pluripotent stem cells (iPSC) may be beneficial for studying disease pathways in the target tissue involved. However, this is an expensive and laborious process, making such cells a suboptimal biomarker cell model. To date, little has been done on patient-derived iPSC RGCs [189]”

Please qualify the statement made on lines 75-77. “…patients with NTG and high-tension glaucoma (HTG) still go blind despite treatment [9]”. Most patients with glaucoma are not blind and will not go blind within their lifetime. Most glaucoma blindness is attributable to late diagnosis. Bilateral blindness is much less common than unilateral blindness. The scenario in which a diagnosed patient is under care and progresses to blindness despite treatment is the exception rather than the rule. Please see: https://pubmed.ncbi.nlm.nih.gov/12689894/ https://pubmed.ncbi.nlm.nih.gov/26302445/ While I am all for making the case that new treatments for glaucoma are urgently needed, it is important not to overstate the risk of blindness for patients already under care. Simply rephrasing to something like “an important subset of patients with NTG…still lose significant vision despite treatment” seems reasonable.

 Agreed. We have changed the text to read: “At present all current treatments for glaucoma are for IOP-lowering and, while lowering IOP can be beneficial to slow progression, an important subset of patients with NTG and high tension glaucoma (HTG) still  lose significant vision  despite treatment [9].”

Minor points:

The point about giant mitochondria in optic nerve head astrocytes is redundantly made twice on lines 87 and 101.

Corrected in text.

Line 99: “neurons” is misspelled.

Corrected in text.

Reviewer 2 Report

The manuscript is thorough and exceptionally well-written on a an interesting topic. Hopefully the review will open a new avenue for clinical studies as we have not seen anything new within glaucoma research for decades.

One minor comment: Tables 1. and 2. To my mind, the findings should be given in chronological order.

Author Response

The manuscript is thorough and exceptionally well-written on an interesting topic. Hopefully the review will open a new avenue for clinical studies as we have not seen anything new within glaucoma research for decades.

The Authors would like to thank the Reviewer for their comments with which to improve the manuscript. 

One minor comment: Tables 1. and 2. To my mind, the findings should be given in chronological order.

This is a good suggestion that we will keep in mind for future papers, however for the purpose of this review we feel that rearranging the findings in chronological order will be tricky. This is because certain findings have been reported by several studies conducted in different times (years). As such, this will result in a much larger table with repeated findings. We therefore felt the table would be easier to read if studies that report each finding are grouped together.